# Scaleable input gradient regularization for adversarial robustness

## Abstract

In this work we revisit gradient regularization for adversarial robustness with some new ingredients. First, we derive new per-image theoretical robustness bounds based on local gradient information. These bounds strongly motivate input gradient regularization. Second, we implement a scaleable version of input gradient regularization which avoids double backpropagation: adversarially robust ImageNet models are trained in 33 hours on four consumer grade GPUs. Finally, we show experimentally and through theoretical certification that input gradient regularization is competitive with adversarial training. Moreover we demonstrate that gradient regularization does not lead to gradient obfuscation or gradient masking.

## 1 Introduction

Neural networks are vulnerable to *adversarial attacks*. These are small (imperceptible to the human eye) perturbations of an image which cause a network to misclassify the image (Biggio et al., 2013; Szegedy et al., 2013; Goodfellow et al., 2014). The threat posed by adversarial attacks must be addressed before these methods can be deployed in error-sensitive and security-based applications (Potember, 2017).

Building adversarially robust models is an optimization problem with two objectives: (i) maintain test accuracy on clean unperturbed images, and (ii) be robust to large adversarial perturbations. The present state-of-the-art method for adversarial defence, adversarial training (Szegedy et al., 2013; Goodfellow et al., 2014; Tramèr et al., 2018; Madry et al., 2017; Miyato et al., 2018), in which models are trained on perturbed images, offers robustness at the expense of test accuracy (Tsipras et al., 2018). Up until the recent method of Shafahi et al. (2019b), multi-step adversarial training had taken many days to train on large datasets (Kannan et al., 2018; Xie et al., 2018).

Assessing the *empirical* effectiveness of an adversarial defence requires careful testing with multiple attacks (Goodfellow et al., 2018). Furthermore, existing defences are vulnerable to new, stronger attacks: Carlini & Wagner (2017a) and Athalye et al. (2018) advocate designing specialized attacks to circumvent prior defences, while Uesato et al. (2018) warn against using weak attacks to evaluate robustness. This has led the community to develop *theoretical* tools to certify adversarial robustness. Several certification approaches have been proposed: through linear programming (Wong & Kolter, 2018; Wong et al., 2018) or mixed-integer linear-programming (Xiao et al., 2018); semi-definite relaxation (Raghunathan et al., 2018b;a); randomized smoothing (Li et al., 2018; Cohen et al., 2019); or estimates of the local Lipschitz constant (Hein & Andriushchenko, 2017; Weng et al., 2018; Tsuzuku et al., 2018). The latter two approaches have scaled well to ImageNet-1k. Notably, randomized smoothing with adversarial training has shown great promise for rigorous certification against attacks in the $\ell_2$ norm (Salman et al., 2019), although it is not yet clear how randomized smoothing may be adapted to other norms.

In practice, certifiably robust networks often perform worse than adversarially trained models, which in general lack theoretical guarantees. In this article, we work towards bridging the gap between theoretically robust networks and empirically effective training methods. Our approach relies on minimizing a loss regularized against large input gradients

$$\mathbb{E}_{(x,y)\sim\mathbb{P}}\left[\mathcal{L}(f(x;w),y) + \frac{\lambda}{2}\|\nabla_x \mathcal{L}(f(x;w),y)\|_*^2\right]$$

where $\| \cdot \|_*$ is dual to the one measuring adversarial attacks (for example the $\ell_1$ norm for attacks measured in the $\ell_\infty$ norm). Heuristically, making loss gradients small should make gradient based attacks more challenging.

Drucker & LeCun (1991) implemented gradient regularization using 'double backpropagation', which has been shown to improve model generalization (Novak et al., 2018). It has been used to improve the stability of GANs (Roth et al., 2017; Nagarajan & Kolter, 2017) and to promote learning robust features with contractive auto-encoders (Rifai et al., 2011). While it has been proposed for adversarial attacks robustness (Lyu et al., 2015; Ross & Doshi-Velez, 2018; Roth et al., 2018; Ororbia II et al., 2017; Hein & Andriushchenko, 2017; Jakubovitz & Giryes, 2018; Simon-Gabriel et al., 2018), experimental evidence has been mixed, in particular, input gradient regularization has so far not been competitive with multi-step adversarial training.

In particular, the heuristic that "gradients should be small" has motivated several other approaches to adversarial robustness which were later shown to suffer from problems of *gradient obfuscation* (Athalye et al., 2018). This is the phenomenon whereby gradients are in some sense hidden or minimized, so that gradient-based attacks fail to produce adversarial examples. Due to this phenomenon, the community now greets adversarial robustness methods based on notions of small gradients with a healthy dose of skepticism. It is therefore necessary to demonstrate new methods do not suffer from gradient obfuscation (Carlini et al., 2019), both empirically and theoretically: empirically, by attacking models with gradient-free methods or non-local attacks; and theoretically using tight certification techniques.

Our main contributions in this work are the following. First, we motivate using input gradient regularization *of the loss* by deriving new theoretical robustness bounds. These bounds use the gradient of the loss, and an estimate of the error of the linear approximation of the loss, called the *modulus of continuity*. These theoretical bounds show that small loss gradient and a small modulus of continuity are sufficient conditions for adversarial robustness. Second, we empirically show that input gradient regularization is competitive with standard adversarial training (Madry et al., 2017), at a fraction of the training time. We verify that training with gradient regularization does not lead to gradient obfuscation, by attacking our trained models with gradient-free methods, and through certification bounds. Finally, we scale input gradient regularization to ImageNet-1k (Deng et al., 2009) by using finite differences to estimate the gradient regularization term, rather than double backpropagation (which does not scale). This allows us to train adversarially robust networks on ImageNet-1k with only 50% more training time than that of standard (non robust) training.

## 2 ADVERSARIAL ROBUSTNESS BOUNDS FROM THE LOSS

### 2.1 BACKGROUND

Much effort has been directed towards determining theoretical lower bounds on the minimum sized perturbation necessary to perturb an image so that it is misclassified by a model. One promising approach, proposed by Hein & Andriushchenko (2017) and Weng et al. (2018), and which has scaled well to ImageNet-1k, is to use the Lipschitz constant of the model. In this section, we build upon these ideas: we propose using the Lipschitz constant of a suitable loss, designed to measure classification errors. Additionally, we propose a tighter lower bound which uses the *local* gradient information of the loss, and an error bound on the first-order approximation of the loss.

Our notation is as follows. Write $y = f(x; w)$ for a model which takes input vectors $x$ to label probabilities, with parameters $w$. Let $\mathcal{L}(y_1, y_2)$ be the loss and write $\ell(x) := \mathcal{L}(f(x, w), y)$, for the loss of a model $f$.

Finding an adversarial perturbation is interpreted as a global minimization problem: find the closest image to a clean image, in some specified norm, that is is also misclassified by the model

$$\min_v \|v\| \quad \text{subject to } f(x + v) \text{ misclassified} \tag{1}$$

However, (1) is a difficult and costly non-smooth, non-convex optimization problem. Instead, Goodfellow et al. (2014) proposed solving a surrogate problem: find a perturbation $v$ of a clean image $x$ that maximizes the loss, subject to the condition that the perturbation be inside a norm-ball

of radius $\delta$ around the clean image. The surrogate problem is written

$$\max_v \ell(x + v) - c(v); \quad \text{where } c(v) = \begin{cases} 0 & \text{if } \|v\| \leq \delta \\ \infty & \text{otherwise} \end{cases} \tag{2}$$

The hard constraint $c(v)$ forces perturbations to be inside the norm-ball centred at the clean image $x$. Ideally, solutions of this surrogate problem (2) will closely align with solutions of the original more difficult global minimization problem. However, the hard constraint in (2) forces a particular scale: it may miss attacks which would succeed with only a slightly bigger norm. Additionally, the maximization problem (2) does not force misclassification, it only asks that the loss be increased.

The advantage of (2) is that it may be solved with gradient-based methods: present best-practice is to use variants of projected gradient descent (PGD), such as the iterative fast-signed gradient method (Kurakin et al., 2016; Madry et al., 2017) when attacks are measured in the $\ell_\infty$ norm. However, gradient-based methods are not always effective: on non-smooth networks, such as those built of ReLU activation functions, a small gradient does not guarantee that the loss remains small locally. That is, ReLU networks may increase rapidly with a very small perturbation, even when local gradients are small. This deficiency was identified in (Papernot et al., 2016) and expanded on by Athalye et al. (2018). In this scenario, PGD methods will fail to locate these worst-case perturbations, and give a false impression of robustness. Carlini & Wagner (2017b) avoid this scenario by incorporating decision boundary information into the loss; others solve (1) directly (Brendel et al., 2018; Chen & Jordan, 2019; Finlay et al., 2019) using gradient-free methods or non-local attacks.

## 2.2 DERIVATION OF THEORETICAL LOWER BOUNDS

This leads us to consider the following compromise between (1) and (2). Consider the following modification of the Carlini & Wagner (2017b) loss $\ell(x) = \max_{i \neq c} f_i(x) - f_c(x)$, where $c$ is the index of the correct label, and $f_i(x)$ is the model output for the $i$-th label. This loss has the appealing property the sign of the loss determines if the classification is correct. Adversarial attacks are found by minimizing

$$\min_v \|v\| \quad \text{subject to } \ell(x + v) \geq \ell_0 \tag{3}$$

The constant $\ell_0$ determines when classification is incorrect; for the modified Carlini-Wagner loss, $\ell_0 = 0$. Problem (3) is closer to the true problem (1), and will always find an adversarial image. We use (3) to derive theoretical lower bounds on the minimum size perturbation necessary to misclassify an image. Suppose the loss is $L$-Lipschitz with respect to model input. Then we have the estimate

$$\ell(x + v) \leq \ell(x) + L\|v\| \tag{4}$$

Now suppose $v$ *is* adversarial, with minimum adversarial loss $\ell(x + v) = \ell_0$. Then rearranging (4), we obtain the lower bound $\|v\| \geq \frac{1}{L}(\ell_0 - \ell(x))$.

Unfortunately, the Lipschitz constant is a global quantity, and ignores local gradient information; see for example Huster et al. (2018). Thus this bound can be quite poor, even when networks have small Lipschitz constant. A tighter theoretical bound may instead be achieved by using (i) the local gradient, and (ii) a bound on the error of a first order approximation of the loss. At image $x$ and perturbations up to size $\varepsilon$, the error of a first order approximation of the loss is bounded above by the modulus of continuity

$$\omega(\varepsilon) = \sup_{x, \|v\| \leq \varepsilon} \ell(x + v) - [\ell(x) + \langle v, \nabla \ell(x) \rangle] \tag{5}$$

Intuitively, this measures the maximum amount the gradient can change over perturbations of size $\varepsilon$. Note that the quantity is defined for any norm. Using this quantity, for perturbations of size $\varepsilon$ we have the estimate

$$\ell(x + v) \leq \ell(x) + \langle v, \nabla \ell(x) \rangle + \omega(\varepsilon)$$
$$\leq \ell(x) + \|v\| \|\nabla \ell(x)\|_* + \omega(\varepsilon) \tag{6}$$

As before, assume $v$ is adversarial, with minimum adversarial loss $\ell(x + v) = \ell_0$. Rearranging (6), we obtain the lower bound $\|v\| \geq \frac{1}{\|\nabla \ell(x)\|_*}(\ell_0 - \ell(x) - \omega(\varepsilon))$.

*Remark* 2.1. If the model and the loss are twice continuously differentiable, the modulus of continuity can be derived from the maximum curvature. However $\omega(\varepsilon)$ is defined even when the model and the loss are not twice differentiable.

We have proved the following.

**Proposition 2.2.** *Suppose the loss $\ell(x)$ is Lipschitz continuous with respect to model input $x$, with Lipschitz constant $L$. Let $\ell_0$ be such that if $\ell(x) < \ell_0$, the model is always correct. Then a lower bound on the minimum magnitude of perturbation $v$ necessary to adversarially perturb an image $x$ is*

$$\|v\| \geq \frac{\max\{\ell_0 - \ell(x), 0\}}{L} \qquad (L\text{-bound})$$

*Suppose in addition the modulus of continuity $\omega(\varepsilon)$ is defined as in* (5). *Then the minimum adversarial distance is bounded below by $\varepsilon$ provided the following inequality holds*

$$\frac{\ell_0 - \ell(x) - \omega(\varepsilon)}{\|\nabla \ell(x)\|_*} \geq \varepsilon \qquad (\omega\text{-bound})$$

Proposition 2.2 motivates the need for input gradient regularization. The Lipschitz constant $L$ is the maximum gradient norm of *the loss* over all inputs. Therefore ($L$-bound) says that a regularization term encouraging small gradients (and so reducing $L$) should increase the minimum adversarial distance. This aligns with (Hein & Andriushchenko, 2017), who proposed the cross-Lipschitz regularizer, penalizing networks with large Jacobians in order to shrink the Lipschitz constant of *the network*.

However, this is not enough: the gap $\ell_0 - \ell(x)$ must be large as well. Together ($L$-bound) explains one form of 'gradient masking' (Papernot et al., 2017). Shrinking the magnitude of gradients while also closing the gap $\ell_0 - \ell(x)$ effectively does nothing to improve adversarial robustness. For example, in one form of defense distillation, the magnitude of the model Jacobian is reduced by increasing the temperature of the final softmax layer of the network. However, this has the detrimental side-effect of sending the model output to $(\frac{1}{N}, \cdots, \frac{1}{N})$, where $N$ is the number of classes, which effectively shrinks the loss gap to zero. An alternative form sends the temperature of the final softmax layer to zero, which pushes the model output to a one-hot vector; however this sends the Lipschitz constant to $\infty$. In either form, the lower bound provided by Proposition 2.2 approaches zero.

Moreover, ($\omega$-bound) states that even supposing the loss's gradient is small, and the gap $\ell_0 - \ell(x)$ is large, there may still be adversarially vulnerable images nearby due to errors in the first order approximation. Taken together, Proposition 2.2 provides three sufficient conditions for training robust networks: (i) the loss gap $\ell_0 - \ell(x)$ should be large; (ii) the gradients of the loss should be small; and (iii) the error of the first-order approximation, measured by $\omega(\varepsilon)$, should also be small. The first point will be satisfied by default when the loss is minimized. The second point will be satisfied by training with a loss regularized to penalize large input gradients. In Section 4 we provide experimental evidence that the third point is satisfied with input gradient regularization implemented using finite differences.

These robustness bounds are most similar in spirit to Weng et al. (2018), who derive bounds using an estimate of the *local* Lipschitz constant of the model by sampling the model gradient norm locally. Indeed, in practice, $L$ and $\omega(\varepsilon)$ can only be estimated and are not exactly computable on all but the simplest networks. We estimate $L$ and $\omega(\varepsilon)$ using the Extreme Value Theory approach outlined by Weng et al. (2018). To estimate $L$, we sample the maximum gradient norm over batches from the data distribution, and estimate an upper bound by fitting a generalized extreme value (GEV) distribution to these samples through maximum likelihood estimation (MLE). Similarly, we estimate $\omega(\varepsilon)$ by sampling the maximum of (5) over batches of the data distribution and $\|v\| \leq \varepsilon$, and then fitting a GEV using MLE. The values of $L$ and $\omega(\varepsilon)$ used in our bounds are estimated from the fitted GEV distribution with threshold $p = 0.001$. Because these quantities are estimates, in practice the theoretical bounds of Proposition 2.2 can only be implemented approximately, providing a heuristic measure of adversarial robustness. Notably however, these bounds are valid in any norm with corresponding dual. Furthemore, upon estimating $L$ and $\omega(\varepsilon)$, we emphasize that these quantities only require *one gradient evaluation and one model evaluation* to provide an estimate on the robustness of an image.

To date, the theoretical bounds in the Euclidean norm that show the greatest promise have arguably been those obtained through randomized smoothing (Li et al., 2018; Cohen et al., 2019; Salman

et al., 2019). In randomized smoothing, models are trained with inputs perturbed through Gaussian noise; robust prediction is achieved by averaging model predictions over inputs again perturbed with Gaussian noise, which requires many tens of thousands of model evaluations. It is well known that training with Gaussian noise is equivalent to squared $\ell_2$ norm gradient regularization (Bishop, 1995). Therefore we expect that models trained with squared norm gradient regularization, should have similar adversarial robustness as those trained through randomized smoothing.

## 3 SQUARED NORM GRADIENT REGULARIZATION

Proposition 2.2 provides strong motivation for input gradient regularization as a method for promoting adversarial robustness. However, it does not tell us what form the gradient regularization term should take. In this section, we show how norm squared gradient regularization arises from a *quadratic cost*.

In adversarial training, solutions of (2) are used to generate images on which the network is trained. In effect, adversarial training seeks a solution of the minimax problem

$$\min_w \mathbb{E}_{x \sim \mathbb{P}} \Big[ \max_v \ell(x + v; w) - c(v) \Big] \tag{7}$$

where $\mathbb{P}$ is the distribution of images. This is a robust optimization problem (Wald, 1945; Rousseeuw & Leroy, 1987). The cost function $c(v)$ penalizes perturbed images from being too far from the original. When the cost function is the hard constraint from (2), perturbations must be inside a norm ball of radius $\delta$. This leads to adversarial training with PGD (Kurakin et al., 2016; Madry et al., 2017). However this forces a particular scale: it is possible that no images are adversarial within radius $\delta$, but that there are adversarial images with only a slightly larger distance. Instead of using a hard constraint, we can relax the cost function to be the quadratic cost $c(v) = \frac{1}{2\delta}\|v\|^2$. The quadratic cost allows attacks to be of any size, but penalizes larger attacks more than smaller attacks. With a quadratic cost, there is less of a danger that a local attack will be overlooked.

Solving (7) directly is expensive: on ImageNet-1k, both Kannan et al. (2018) and Xie et al. (2018) required large-scale distributed training with many dozens or hundreds of GPUs, and over a week of training time. Recent research suggests this hurdle may be surmounted (Shafahi et al., 2019b; Zhang et al., 2019). However we take the view that (7) may be bounded above, and solved approximately. When the loss is smooth, the modulus of continuity is $\omega(\varepsilon) = \varepsilon^2 C$, where $C$ is the maximum positive Hessian norm over all inputs. By using the bound (6) with the quadratic loss $c(v) = \frac{1}{2\delta}\|v\|^2$, the optimal value of $\max_v \ell(x + v) - c(v)$ is $\frac{\delta}{2(1-\delta C)}\|\nabla_x \ell(x)\|_*^2$, provided $\delta < \frac{1}{C}$. This gives the following proposition.

**Proposition 3.1.** *Suppose both the model and the loss are twice continuously differentiable. Suppose attacks are measured with quadratic cost $\frac{1}{2\delta}\|v\|^2$. Then the optimal value of* (7) *is bounded above by*

$$\min_w \mathbb{E}_{x \sim \mathbb{P}} \left[ \ell(x; w) + \frac{\lambda}{2}\|\nabla_x \ell(x)\|_*^2 \right] \tag{8}$$

*where $\lambda = \frac{\delta}{1-\delta C}$.*

That is, we may bound the solution of the adversarial training problem (7) by solving the gradient regularization problem (8), when the cost function is quadratic. It is not necessary to know $\delta$ or compute $C$; they are absorbed into $\lambda$.

Input gradient regularization using the squared $\ell_2$ norm was proposed for adversarial robustness in Ross & Doshi-Velez (2018). It was expanded by Roth et al. (2018) to use a Mahalanobis norm with the correlation matrix of adversarial attacks.

When $c(v)$ is the hard constraint forcing attacks inside the $\delta$ norm ball and $C$ is small, supposing the curvature term is negligible, we can estimate the maximum in (7) by $\ell(x) + \frac{1}{\delta}\|\nabla_x \ell(x)\|_*$, using the dual norm for the gradient. This is norm gradient regularization (not squared), and has been used for adversarial robustness on both CIFAR-10 (Simon-Gabriel et al., 2018), and MNIST (Seck et al., 2019).

### 3.1 FINITE DIFFERENCE IMPLEMENTATION

Norm squared input gradient regularization has long been used as a regularizer in neural networks: Drucker & LeCun (1991) first showed its effectiveness for generalization. Drucker & LeCun

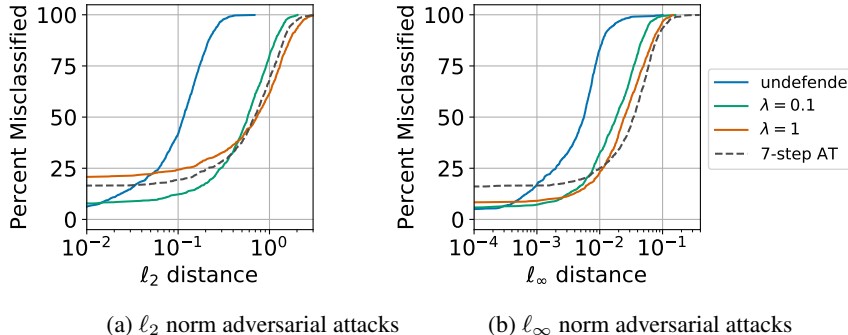

(a) $\ell_2$ norm adversarial attacks    (b) $\ell_\infty$ norm adversarial attacks

Figure 1: Adversarial attacks on the CIFAR-10 dataset, on networks built with standard ReLUs. Regularized networks attacked in $\ell_2$ are trained with squared $\ell_2$ norm gradient regularization; networks attacked in $\ell_\infty$ are trained with squared $\ell_1$ norm regularization.

implemented gradient regularization with 'double backpropagation' to compute the derivatives of the penalty term with respect to the model parameters $w$, which is needed to update the parameters during training. Double backpropagation involves two passes of automatic differentiation: one pass to compute the gradient of the loss with respect to the inputs $x$, and another pass on the output of the first to compute the gradient of the penalty term with respect to model parameters $w$. In neural networks, double backpropagation is the standard technique for computing the parameter gradient of a regularized loss. However, it is not currently scaleable to large neural networks. Instead we approximate the gradient regularization term with finite differences.

**Proposition 3.2** (Finite difference approximation of squared $\ell_2$ gradient norm)**.** *Let $d$ be the normalized input gradient direction: $d = \nabla_x \ell(x)/\| \nabla_x \ell(x)\|_2$ when the gradient is nonzero, and set $d = 0$ otherwise. Let $h$ be the finite difference step size. Then, the squared $\ell_2$ gradient norm is approximated by*

$$\| \nabla_x \ell(x)\|_2^2 \approx \left( \frac{\ell(x + hd) - \ell(x)}{h} \right)^2 \tag{9}$$

The vector $d$ is normalized to ensure the accuracy of the finite difference approximation: as can be seen by (6), the error of the finite difference is bounded by $\omega(h)/h$. Without normalizing the gradient, this term can be large, which is undesirable in general. However, the finite difference error term does play a role in promoting adversarial robustness. Proposition 2.2 shows it is necessary for $\omega(\varepsilon)$ to be small. By using finite differences, minimizing (8) also implicitly minimizes $\omega(h)$, due to the error term. Therefore the finite difference approximation provide an indirect means for controlling the modulus of continuity $\omega(\varepsilon)$ necessary for adversarial robustness, through the finite difference error. This would otherwise be unavailable with an exact method computing the gradient norm.

The finite differences approximation (9) allows the computation of the gradient of the regularizer (with respect to model parameters $w$) to be done with only two regular passes of backpropagation, rather than with double backpropagation. On the first, the input gradient direction $d$ is calculated. The second computes the gradient with respect to model parameters by performing backpropagation on the right-hand-side of (9). Double backpropagation is avoided by detaching $d$ from the computational graph after the first pass. In practice, for large networks, we have found that the finite difference approximation of the regularization term is considerably more efficient than using double backpropagation. On networks sized for MNIST, we have observed roughly a 10% speed improvement by using finite differences; on CIFAR-10 networks finite differences are roughly 50% faster than double backpropagation.

Our proposed training algorithm, with squared Euclidean input gradient regularization, is presented in Algorithm 1 of Appendix A. Other gradient penalty terms can be approximated as well. For example, when defending against attacks measured in the $\ell_\infty$ norm, the squared $\ell_1$ norm penalty can approximated by setting instead $d = \text{sign}(\nabla_x \ell(x))/\sqrt{N}$ when the gradient is nonzero.

Table 1: Adversarial robustness statistics, measured in the $\ell_\infty$ norm. Top1 error is reported on CIFAR-10; Top5 error on ImageNet-1k. On CIFAR-10 gradient regularization is reported using either double backpropagation (DBP) or finite differences (FD).

| | % clean error | $\omega$-bound % error at | | empirical % error at | | training time (hours) |
|---|---|---|---|---|---|---|
| | | $\varepsilon = \frac{2}{255}$ | $\varepsilon = \frac{8}{255}$ | $\varepsilon = \frac{2}{255}$ | $\varepsilon = \frac{8}{255}$ | |
| **CIFAR-10** | | | | | | |
| Undefended | **4.36** | 100 | 100 | 70.82 | 98.94 | 2.06 |
| $\ell_\infty$ 7-step AT, $\varepsilon = 2/255$ | 6.17 | 16.02 | 100 | **16.80** | 53.90 | 10.82 |
| $\ell_\infty$ 7-step AT, $\varepsilon = 8/255$ | 16.33 | 26.59 | 58.32 | 22.86 | **46.02** | 12.10 |
| $\ell_\infty$ 'For Free' AT, $m = 2$ | 6.82 | 20.08 | 100 | 22.61 | 63.95 | 1.91 |
| $\ell_\infty$ 'For Free' AT, $m = 8$ | 19.94 | 38.52 | 100 | 34.09 | 61.48 | 2.04 |
| squared $\ell_1$ norm (DBP), $\lambda = 0.1$ | 6.50 | 25.69 | 100 | 41.63 | 97.31 | 9.85 |
| squared $\ell_1$ norm (DBP), $\lambda = 1$ | 12.62 | 33.05 | 99.43 | 33.95 | 80.37 | 9.89 |
| squared $\ell_1$ norm (FD), $\lambda = 0.1$ | 6.45 | 19.40 | 100 | 24.92 | 70.41 | 5.22 |
| squared $\ell_1$ norm (FD), $\lambda = 1$, $h = 0.1$ | 9.02 | 20.66 | 55.30 | 18.47 | 58.69 | 5.15 |
| **ImageNet-1k** | | | | | | |
| Undefended | **6.94** | 100 | 100 | 90.21 | 98.94 | 20.30 |
| squared $\ell_2$ norm, $\lambda = 0.1$ | 7.66 | 74.7 | 100 | 70.56 | 97.53 | 32.60 |
| squared $\ell_2$ norm, $\lambda = 1$ | 10.26 | 63.7 | 100 | **52.79** | **95.93** | 33.87 |

## 4 EXPERIMENTAL RESULTS

In this section we provide empirical evidence that input gradient regularization is an effective tool for promoting adversarial robustness. We attack networks with gradient-free attacks, and demonstrate gradient regularization does not lead to gradient obfuscation. Moreover our results show that gradient regularization is comparable to adversarial training, both experimentally and through rigorous certification.

We train networks on the CIFAR-10 dataset (Krizhevsky & Hinton, 2009), and ImageNet-1k (Deng et al., 2009). On the CIFAR dataset we use the ResNeXt-34 (2x32) architecture; on ImageNet-1k we use a ResNet-50 (He et al., 2016). The CIFAR networks were trained with standard data augmentation and learning rate schedules on a single GeForce GTI 1080 Ti. On ImageNet-1k, we modified the training code of Shaw et al.'s [44] submission to the DAWNBench competition (Coleman et al., 2018) and train with four GPUs. Training code and trained model weights will be made available.

We train an undefended network as a baseline to compare various types of regularization. On CIFAR-10, networks are trained with squared $\ell_2$ and squared $\ell_1$ gradient norm regularization. The former is appropriate for defending against attacks measured in $\ell_2$; the latter for attacks measured in $\ell_\infty$. We set the regularization strength to be either $\lambda = 0.1$ or 1; and set finite difference discretization $h = 0.01$ unless otherwise specified. (For a study of the effect of finite difference size $h$, refer to Table 4.) We compare each network with the most common form of adversarial training, where models are trained using the hyperparameters in Madry et al. (2017) (7-steps of FGSM, projected onto an $\ell_\infty$ ball of either radius $\frac{8}{255}$ or $\frac{2}{255}$). On CIFAR-10 we also compare against the 'For Free' implementation of Shafahi et al. (2019a), using the same hyperparameters. On ImageNet-1k we only train adversarially robust models with squared $\ell_2$ regularization.

On each dataset, we attack 1000 randomly selected images. We perturb each image with attacks in both the Euclidean and $\ell_\infty$ norms, with a suite of current state-of-the-art attacks: the Carlini-Wagner attack (Carlini & Wagner, 2017b); the Boundary attack (Brendel et al., 2018) (a gradient-free attack); the LogBarrier attack (Finlay et al., 2019) (a non-local attack); and PGD (Madry et al., 2017) (in both the $\ell_\infty$ norm or the $\ell_2$ norm). The former three attacks have all been shown effective at evading gradient masking and gradient obfuscation. PGD excels at finding images close to the original when gradients are not close to zero. We report our results using the best adversarial distance *on a per image basis*, in each norm. On the majority of test images, PGD finds the closest adversarial image. However on a significant portion of inputs, the closest adversarial image is found by one of the three other attacks.

Table 2: Adversarial robustness statistics, measured in $\ell_2$. Top1 error is reported on CIFAR-10; Top5 error on ImageNet-1k. On CIFAR-10 gradient regularization is reported using either both double backpropagation (DBP) or finite differences (FD).

| | % clean error | mean adversarial distance | | | training time (hours) |
|---|---|---|---|---|---|
| | | $L$-bound | $\omega$-bound | empirical | |
| **CIFAR-10** | | | | | |
| Undefended | **4.36** | 0.006 | 0 | 0.12 | 2.06 |
| $\ell_\infty$ 7-step AT, $\varepsilon = 2/255$ | 6.17 | 0.05 | 0.36 | 0.66 | 10.82 |
| $\ell_\infty$ 7-step AT, $\varepsilon = 8/255$ | 16.33 | 0.18 | **0.55** | 0.74 | 12.10 |
| $\ell_\infty$ 'For Free' AT, $m = 2$, $\lambda = 0.1$ | 6.82 | 0.06 | 0.35 | 0.53 | 1.91 |
| $\ell_\infty$ 'For Free' AT, $m = 8$, $\lambda = 1$ | 19.94 | 0.03 | 0.29 | 0.49 | 2.04 |
| squared $\ell_2$ norm (DBP), $\lambda = 0.1$ | 5.91 | 0.006 | 0 | 0.13 | 9.60 |
| squared $\ell_2$ norm (DBP), $\lambda = 1$ | 4.35 | 0.01 | 0 | 0.15 | 9.74 |
| squared $\ell_2$ norm (FD), $\lambda = 0.1$ | 8.03 | 0.14 | 0.34 | 0.63 | 5.18 |
| squared $\ell_2$ norm (FD), $\lambda = 1$ | 20.31 | **0.30** | 0.50 | **0.81** | 5.08 |
| **ImageNet-1k** | | | | | |
| Undefended | **6.94** | $3.63\mathrm{e}{-2}$ | 0.5 | 0.55 | 20.30 |
| squared $\ell_2$ norm, $\lambda = 0.1$ | 7.66 | 0.13 | 0.96 | 1.14 | 32.60 |
| squared $\ell_2$ norm, $\lambda = 1$ | 10.26 | **0.26** | **1.23** | **1.75** | 33.87 |

Adversarial robustness results for networks attacked in the $\ell_\infty$ norm are presented in Table 1. These results are for networks built of standard ReLUs. Table 1 and Figure 1 demonstrate a clear trade-off between test accuracy and adversarial robustness, as the strength of the regularization is increased. On CIFAR-10, the undefended network achieves test error of 4.36%, but is not robust to attacks even at $\ell_\infty$ distance $\frac{2}{255}$. However with a strong regularization parameter ($\lambda = 1$), test error increases to 9.02% on clean images. At both adversarial distance $\varepsilon = \frac{2}{255}$ and $\frac{8}{255}$ adversarial training achieves slightly better robustness (by about 5% percent) than our best models, yet comes at over double the training time.

In Table 1 we also report heuristic measures of robustness using ($\omega$-bound). We skip reporting bounds using ($L$-bound), for this bound returned vacuous results (an upper bound of 100% test error at all distances). On some test images the heuristic bound fails; this is due to the inherent difficulty of estimating $\omega(\varepsilon)$ accurately.

In Table 2 we report results on models trained for attacks in the $\ell_2$ norm. On CIFAR-10, the most robust model is trained with regularization strength $\lambda = 1$, and outperforms even the adversarially trained model. On ImageNet-1k, we see the same pattern: the model trained with $\lambda = 1$ offers the best protection against adversarial attacks. Due to the long training time, we were not able to train ImageNet-1k with standard multi-step adversarial training. In Table 2 we also report our heuristic measure of bounds on the minimum distance required to adversarially perturb, using the Carlini-Wagner loss.[1] Further $\omega$-bound results on CIFAR-10 are presented in Table 6 Both ($L$-bound) and ($\omega$-bound) provide reasonable estimates of the minimum adversarial distance, though ($\omega$-bound) is closer to empirical results. These heuristic measures of robustness are further corroborated by Table 3 of Appendix B, which shows that both adversarial training and gradient regularization significantly decrease model gradients and the modulus of continuity $\omega(\varepsilon)$. We also use randomized smoothing, a tight and exact bound on robustness, to verify our heuristic measures, see Table 5.

## 5 CONCLUSION

We have provided motivation for training adversarially robust networks through input gradient regularization, by bounding the minimum adversarial distance with gradient statistics of the loss. We have shown that gradient regularization is scaleable to ImageNet-1k, and provides adversarial robustness competitive with adversarial training, both theoretically and empirically. Moreover we have demonstrated that gradient regularization does not lead to gradient obfuscation. Networks

---

[1]This loss can be modified for Top-5 mis-classification as well.

may be trained using gradient regularization in fractions of the training time of standard adversarial training.

We gave theoretical per-image bounds on the minimum adversarial distance, for non-smooth models, using the Lipschitz constant of the loss, and its modulus of continuity, which measures the error of a first order approximation of the loss. These bounds were empirically compared against state-of-the-art attacks and other theoretical bounds. Although in practice these bounds give heuristic measures of robustness, they are valid in any norm, and provide per-image robustness measures at the cost of only one model evaluation and one gradient evaluation.

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

## A    TRAINING ALGORITHM

---

**Algorithm 1** Training with squared $\ell_2$-norm input gradient regularization, using finite differences

---

1: **Input:** Initial model parameters $w_0$
   **Hyperparameters:** Regularization strength $\lambda$; batch size $m$; finite difference discretization $h$
2: **while** $w_t$ not converged **do**
3:    sample minibatch of data $\left\{(x^{(i)}, y^{(i)})\right\}_{i=1,\dots,m}$ from empirical distribution $\hat{\mathbb{P}}$
4:    **for** $i = 0$ **to** $m$ **do**
5:        $g^{(i)} = \nabla_x \ell(x^{(i)}, y^{(i)}; w_t)$
6:        $d^{(i)} = \begin{cases} \frac{g^{(i)}}{\|g^{(i)}\|_2} & \text{if } g^{(i)} \neq 0 \\ 0 & \text{otherwise} \end{cases}$     $\triangleright$ for $\ell_1$-norm use normalized signed gradient
7:        detach $d^{(i)}$ from computational graph
8:        $z^{(i)} = x^{(i)} + h d^{(i)}$
9:    **end for**
10:    $\mathcal{L}(w) = \frac{1}{m} \sum_{i=1}^{m} \ell(x^{(i)}, y^{(i)}; w)$
11:    $\mathcal{R}(w) = \frac{1}{m} \sum_{i=1}^{m} \frac{1}{h^2} \left( \ell(z^{(i)}, y^{(i)}; w) - \ell(x^{(i)}, y^{(i)}; w) \right)^2$
12:    $w_{t+1} \leftarrow w_t - \tau_t \nabla_w \left( \mathcal{L}(w_t) + \lambda \mathcal{R}(w_t) \right)$
13: **end while**

---

## B    ADDITIONAL EMPIRICAL RESULTS

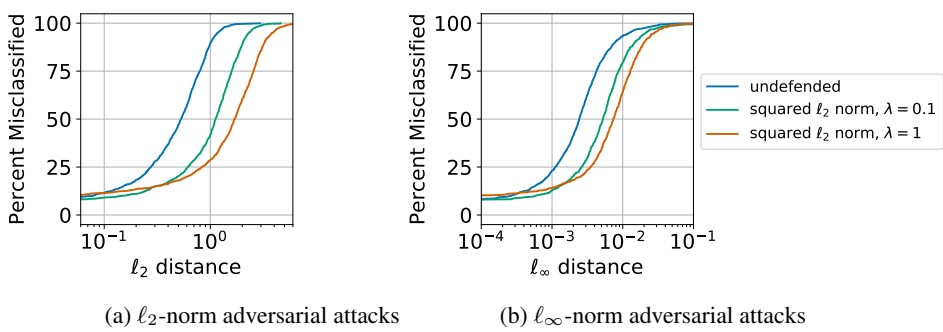

(a) $\ell_2$-norm adversarial attacks      (b) $\ell_\infty$-norm adversarial attacks

Figure 2: Adversarial attacks on ImageNet-1k with the ResNet-50 architecture. Top5 error reported.

Table 3: Regularity statistics on selected models, measured in the $\ell_2$ norm. Statistics computed using modified loss $\max_{i \neq c} f_i(x) - f_c(x)$.

| | $\|\nabla \ell(x)\|$ | | $\omega(0.5)$ | $\omega(1.0)$ |
| | mean | max | | |
| --- | --- | --- | --- | --- |
| **CIFAR-10** | | | | |
| Undefended | 3.05 | 122.34 | 1.80 | 3.13 |
| $\ell_\infty$ 7-step AT, $\varepsilon = 2/255$ | 0.63 | 11.96 | 0.08 | 0.32 |
| $\ell_\infty$ 7-step AT, $\varepsilon = 8/255$ | 0.40 | 2.52 | 0.02 | 0.06 |
| $\ell_\infty$ 'For Free', $m = 2$ | 0.75 | 10.82 | 0.13 | 0.35 |
| $\ell_\infty$ 'For Free', $m = 8$ | 0.93 | 16.08 | 0.11 | 0.31 |
| squared $\ell_2$ norm (DBP), $\lambda = 0.1$ | 3.38 | 107.13 | 1.60 | 1.40 |
| squared $\ell_2$ norm (DBP), $\lambda = 1$ | 2.75 | 56.09 | 1.31 | 2.30 |
| squared $\ell_2$ norm (FD), $\lambda = 0.1$ | 0.58 | 4.43 | 0.08 | 0.50 |
| squared $\ell_2$ norm (FD), $\lambda = 1$ | 0.35 | 1.33 | 0.02 | 0.05 |
| **ImageNet-1k** | | | | |
| Undefended | 1.12 | 17.51 | 1.7e−2 | 6.7e−2 |
| squared $\ell_2$ norm, $\lambda = 0.1$ | 0.46 | 4.85 | 5.5e−3 | 2.2e−2 |
| squared $\ell_2$ norm, $\lambda = 1$ | 0.27 | 2.12 | 3.4e−3 | 2.1e−2 |

Table 4: Study of empirical robustness in the $\ell_\infty$ norm as a function of finite difference size $h$. All models here trained with squared $\ell_1$ norm gradient penalty, with regularizer strength $\lambda = 1$.

| | % clean error | empirical % error at | |
| | | $\varepsilon = \frac{2}{255}$ | $\varepsilon = \frac{8}{255}$ |
| --- | --- | --- | --- |
| Undefended | 4.36 | 70.82 | 98.94 |
| double backpropagation | 12.62 | 33.95 | 80.37 |
| $h = 1e{-}4$ | 39.68 | 50.95 | 84.20 |
| $h = 1e{-}3$ | 25.84 | 38.81 | 78.65 |
| $h = 1e{-}2$ | 14.47 | 26.54 | 69.91 |
| $h = 0.1$ | 9.02 | 18.47 | 58.69 |
| $h = 1$ | 5.51 | 45.93 | 83.86 |

Table 5: Certified test error (%) at various $\ell_2$ radii on CIFAR-10, using the randomized smoothing certification technique of Cohen et al. (2019).

| | 0.25 | 0.5 | 0.75 | 1.0 | 1.25 | 1.5 | 1.75 | 2.0 |
| --- | --- | --- | --- | --- | --- | --- | --- | --- |
| Undefended | 90 | 92 | 96 | 99 | 100 | 100 | 100 | 100 |
| squared $\ell_2$ norm, $\lambda = 1$ | 47 | 63 | 80 | 90 | 90 | 90 | 90 | 90 |
| Cohen et al. (2019) | 40 | 57 | 68 | 77 | 83 | 86 | 88 | 90 |
| Salman et al. (2019) | 26 | 43 | 52 | 62 | 67 | 71 | 75 | 81 |

Table 6: Certified test error (%) at various $\ell_2$ radii on CIFAR-10, using ($\omega$-bound).

| | 0.25 | 0.5 | 0.75 | 1.0 | 1.25 | 1.5 |
| --- | --- | --- | --- | --- | --- | --- |
| Undefended | 100 | 100 | 100 | 100 | 100 | 100 |
| $\ell_\infty$ 7-step AT, $\varepsilon = 2/255$ | 19 | 91 | 100 | 100 | 100 | 100 |
| $\ell_\infty$ 7-step AT, $\varepsilon = 8/255$ | 31 | 47 | 61 | 77 | 100 | 100 |
| squared $\ell_2$ norm (FD), $\lambda = 0.1$ | 22 | 99 | 100 | 100 | 100 | 100 |
| squared $\ell_2$ norm (FD), $\lambda = 1$ | 33 | 50 | 65 | 80 | 98 | 100 |

