# OpenReview forum: "Scaleable input gradient regularization for adversarial robustness"
_ICLR.cc/2020/Conference — Reject_

### Official Review · AnonReviewer3 · 2019-10-15
**Official Blind Review #3**

**Rating:** 6

**Review:**

Summary:
This paper provides new understandings on adversarial robustness from the perspective of input gradient regularization. Input gradient regularization hasn't been able to achieve comparable robustness to  adversarial training. Built upon existing works, this paper derives two minimum perturbation bounds (L-bound and w-bound) to explain this, as well as other defenses such as Lipschitz regularization and defensive distillation. Taking a step further, this paper proposes to use the finite difference to estimate the input gradients, which not only gives a nice property for reduced modulus of continuity (eg. the w-bound), but also makes the regulation scalable to large networks and datasets. I quite like the theoretical connections derived in this paper. Empirical evidences support their claims, and demonstrate indeed comparable robustness of input gradient regularization to adversarial training.

The empirical results can be strengthened by including the normal input gradient regularization baseline (using double backpropagation), at least on cifar-10. This is less likely to change the conclusions, but  would be interesting to see the comparisons.

Note that, there are already new progresses in adversarial training:
[1] Wang, Yisen, et al. "On the Convergence and Robustness of Adversarial Training." ICML, 2019.
[2] Zhang, Hongyang, et al. "Theoretically principled trade-off between robustness and accuracy." ICML, 2019.
[3] Carmon, Yair, et al. "Unlabeled data improves adversarial robustness." NeurIPS, 2019.
[4] Uesato, et al. "Are Labels Required for Improving Adversarial Robustness?" NeurIPS, 2019.

==========
After rebuttal:
Thanks for the new results. My rating remains the same.

**Experience Assessment:**

I have published one or two papers in this area.

**Review Assessment: Checking Correctness Of Derivations And Theory:**

I carefully checked the derivations and theory.

**Review Assessment: Checking Correctness Of Experiments:**

I carefully checked the experiments.

**Review Assessment: Thoroughness In Paper Reading:**

I read the paper thoroughly.

---

> ### Author Response · Authors · 2019-11-09
> **Reply to reviewer 3**
>
> Thank you for review, and for the referrals to new results.
>
> Following your and Reviewer 2’s comments, we have updated the empirical results to compare against double backpropagation (DBP). DBP does offer some improvement in adversarial robustness, although not as significant as with finite differences. We hope that it is clear that the use of finite differences is important for controlling the modulus of continuity, and so keeping (w)-bound small. Moreover, on CIFAR-10 at least, finite differences are nearly twice as fast as DBP. Due to computational constraints we have not attempted DBP training on ImageNet-1k.

---

### Official Review · AnonReviewer2 · 2019-10-21
**Official Blind Review #2**

**Rating:** 3

**Review:**

This paper revisits the method of gradient regularization, which regularizes the loss function by adding the norm of the input gradient, aiming to improve adversarial robustness. The standard gradient regularization is implemented with "double backpropagation", which could be time consuming for large networks. This paper proposes to replace the exact gradient by its discretization via finite difference, which is computationally more efficient comparing to "double backpropagation".  Experiments are conducted to show the effectiveness of the method in reducing training time.

Overall, the paper is well written but the contribution is very limited and I find some of the experimental comparisons unfair.  Thus I do not support publication of the paper. Here are my detailed arguments.

1) The gradient regularization is not novel
The main contribution of the paper is to use the finite difference in the gradient regularization in order to improve the training time.  The method of gradient regularization is not new, hence the contribution is only the computational effectiveness. This would be fine if one could improve the state-of-the-art method's training time by a lot, but according to the experiments, the performance of adversarial robustness is far from the adversarial training, for example in CIFAR10 epsilon=8/255, there is a 12% drop, which is a huge gap.

2) Unfair comparison in the experiments
In table 2, the performance of robustness with respect to the l2 norm is presented. However, in the baseline method of adversarial training, the used attack is in l_{infty} norm. This is unfair since l2 norm and l_{infty} norm has very different characteristics. Moreover, the gradient regularization uses l2 norm instead of l_{1} as in table 1, which makes the comparison unfair.

Furthermore, the finite difference improves the training time versus the standard gradient regularization, but it does not imply that the performance will be the same. In particular, it would be better to include the performance of standard gradient regularization in table 1 and 2 as well.

3) Comment on the motivation/theory
The theory part is fairly straightforward and it clearly shows that the norm of gradient (w.r.t input) itself is not sufficient to guarantee robustness. As a evidence, even under standard training (for example on MNIST), the gradient norm could be very small, in order of 10^{-4} but still have adversarial example with very small perturbation.  Thus, what we also need is to control how fast this gradient changes (Lipschitz constant or w-bound). However, the gradient norm regularization does not take into account how gradient changes. It is claimed that the finite discretization implicitly reduces how the gradient changes, however, I am not convinced by the argument since as long as we take h to be small, it is still a very local measure. An interesting question would be how h affect the performance and it is not discussed in the paper.

**Experience Assessment:**

I have read many papers in this area.

**Review Assessment: Checking Correctness Of Derivations And Theory:**

I assessed the sensibility of the derivations and theory.

**Review Assessment: Checking Correctness Of Experiments:**

I assessed the sensibility of the experiments.

**Review Assessment: Thoroughness In Paper Reading:**

I read the paper thoroughly.

---

> ### Author Response · Authors · 2019-11-09
> **Reply to reviewer 2**
>
> Thanks for taking the time to review our work. We’ll address each criticism directly.
>
> 1) Our main contribution is not gradient regularization and its finite difference implementation, taken separately these are both well-established techniques. Our main contribution is the theoretical justification for gradient regularization, and empirical results showing that gradient regularization is indeed effective for adversarial training. Prior to our work there has been much debate as to whether gradient regularization could be effective for robustness; some had suggested gradient regularization results in gradient obfuscation. We show this is not so when finite differences are used. Moreover finite differences yield significant speed-ups in training time.
>
> Our results show gradient regularization yields comparable adversarial robustness to adversarial training, at a fraction at the training time. That is, their respective robustness results are on the same order. Focusing on one number (L-inf results at \eps=8/255) without considering context is a poor characterization of our work.
>
> We are not claiming gradient regularization is a replacement for adversarial training, or that it is better. However we do show empirically that it can achieve quite good results. Given that our implementation is scaleable, practitioners may want to use gradient regularization when training time is a limiting  factor. This is an important point.
>
> 2) There are many works showing that L-infinity adversarial training provides robustness in L2, and vice versa (see eg Madry et al’s oft-cited work). We chose to compare against L-infinity trained models, because there is a common set of agreed upon hyper-parameters used in the literature for L-infinity training. Other L2 robustness papers have compared against L-infinity trained models. For example Qian and Wegman (2019,  https://openreview.net/forum?id=ByxGSsR9FQ) compare against L-infinity adversarially trained models, even though their results (which at the time were state-of-the-art) are for robustness in L2.
>
> Following your comments and that of Reviewer 3, we have now included comparisons with ‘double backprop’ gradient regularization, updated in Tables 2 and 3 of the revised manuscript.
>
> 3) This is exactly the point we are trying to make: (w)-bound must be small, however controlling the estimate of the gradient (with double backprop) does not suffice to control the modulus of continuity. In contrast, the modulus of continuity is controlled via finite differences. See Table 3, which lists modulus of continuity estimates, and is now updated to compare finite differences  against double backpropagation. As you can see in the updated Tables 1 & 2, double backpropagation yields worse adversarial robustness than with finite differences, and is much slower.
>
> In other words, the error of the finite difference approximation is quite important for controlling the modulus of continuity. It is important not to make h small.
>
> As an aside, since these models are not even differentiable, in the limit as h goes to zero, finite differences do not provide the same gradient estimate as backpropagation.  For illustration, take f(x) = max(x,0) + min(x,0). Finite differences yield f’(0)=1, whereas backpropagation yields f’(0) = 2. Finite differences are analytically correct, but backpropagation is not.

---

> > ### Comment · AnonReviewer2 · 2019-11-09
> > **Re: Reply to reviewer 2**
> >
> > Thank you for the clarification.
> > As far as I understand, the theoretical part does not take into account the finite difference, which is the key for controlling the modulus of continuity in practice. In which case, the theory does not provide enough understanding of the phenomenon and I would not consider the observation as theoretically justified.  Please comment on it.

---

> > > ### Author Response · Authors · 2019-11-11
> > > **Finite differences control modulus of continuity**
> > >
> > > Hi, thank you for the incisive observation. Indeed Section 2 discusses why it is necessary to control the modulus of continuity; not until Section 3.1 do we discuss the finite difference implementation of the gradient penalty term.
> > >
> > > We (briefly) addressed why the finite difference implementation controls the modulus of continuity, in the paragraph following Proposition 3.2. Admittedly we could expand on this paragraph a bit more in the text if so desired, please let us know.
> > >
> > > In detail, one way to see why finite differences control the modulus of continuity as follows. From the fundamental theorem of calculus, we have that for a piecewise differentiable function $f(x)$ (such as the loss of a model with ReLU activations) the finite difference is
> > > $$\frac{f(x+h) - f(x)}{h} = \frac{1}{h} \int_0^1 f'(x+th) dt$$
> > > In other words, the finite difference is the average of the derivative along the line $[x, x+h]$. The modulus of continuity is the maximum amount of change in the derivative. If the modulus of continuity is large, then so too must the average; and vice versa. Thus if we control the average (via finite differences), we also control the maximum change in the derivative (ie, the modulus of continuity).
> > >
> > > [edit] We'd also like to remark that empirically, the finite difference implementation of the gradient penalty term does an excellent job of controlling the modulus of continuity: see Table 3, where we report the modulus of continuity $\omega(0.5)$ and $\omega(1)$. Finite differences is as good as adversarial training in controlling $\omega$.

---

> > > > ### Comment · AnonReviewer2 · 2019-11-14
> > > > **Re: Finite differences control modulus of continuity**
> > > >
> > > > I agree with the author's intuition that finite difference encourages the control of the modulus of continuity. I believe a more rigorous study on it would substantially strengthen the paper (for example assuming the function is twice differentiable and theoretically studies how the modulus of continuity varies).
> > > >
> > > > I thank the authors for providing studies on the affect of h. There is some interesting pattern in this experimental results. In particular, when h is small the clean error is very high, as h increases, the clean error decreases and almost achieve the same performance in standard training. This is somehow counter-intuitive because one may expect double backpropagation matches the small h case. Moreover, up to h = 1e-3, the performance is not as good as double backpropagation in both clean and adv setting. I believe more investigation are needed to better understand the phenomenon.
> > > >
> > > > Overall, the idea has merit and potential, but more rigorous study is needed to understand the phenomenon.

---

> > > > > ### Author Response · Authors · 2019-11-14
> > > > > **More theory and a summary**
> > > > >
> > > > > We can elaborate more on the link between finite differences and the modulus of continuity when the model is twice differentiable. In that case, the modulus of continuity is proportional to the maximum curvature. For illustration, in 1D $\omega(\epsilon) = \sup_{x,v\leq\epsilon} f(x+v) - f(x) - v f’(x)$. Use Taylor’s theorem to expand $f(x+v) = f(x) + v f’(x) + v^2 f{’}{'}(\xi)$ for some $x\leq\xi\leq x+v$, which gives
> > > > >  $$\omega(\epsilon) =  \epsilon^2 \sup_{\xi} f{’}{'}(\xi)$$
> > > > > In other words, the modulus of continuity is $\epsilon^2$ times the maximum curvature. In n-dimensions this can be generalized, so that the modulus of continuity is proportional to the maximum Hessian norm (induced by the particular norm used to measure adversarial robustness), or colloquially the maximum curvature.
> > > > >
> > > > > Now turning to our finite difference regularization term. Again from Taylor’s theorem the finite difference is exactly
> > > > > $$\frac{ f(x+v) – f(x)}{h} = \langle \nabla f(x), v \rangle + h \langle v, \nabla^2 f(\xi) v \rangle $$
> > > > > for some $\xi$ along the line $(x, x+v)$.
> > > > > By squaring this quantity, we control both the gradient and the Hessian in the direction of $v$. When $v=\frac{\nabla f(x)}{||\nabla f(x)||}$, as in our method, we are controlling the gradient norm squared, as well as the curvature (Hessian) in the direction of the gradient.
> > > > >
> > > > > We could formalize this argument in an appendix section if so desired.
> > > > >
> > > > > Note that this squared finite difference term does not control the maximum curvature everywhere (the modulus of continuity), but it does provide some control on the curvature locally. We remark that it is not possible for regularization to control the worst-case curvature everywhere across all inputs, however it does a good job locally and empirically. Just as gradient regularization provides some control of the worst-case gradient (the Lipschitz constant -- see eg Hein  & Andriushchenko, NIPS 2017 or Weng et al, ICLR 2018), our method provides some control over the worst-case curvature (the modulus of continuity).
> > > > >
> > > > > A brief remark on finite difference error and autograd error. In the limit as $h$ goes to zero, finite differences do not in general recover the autograd estimate of the gradient. This is because models are not differentiable. For example take $f(x) = |x| = \max\{x,0\} +\max\{-x,0\}$. At $x=0$ forward finite differences give $f’(0)=1$, whereas autograd yields $f’(0)=0$. We believe this hints at the reason for the counter-intuitive results for small $h$. However the fact that autograd and finite differences do not agree in the limit as $h \rightarrow 0$ is a difficult and involved topic best addressed by others (see for example Bolte and Pauwels 2019, https://arxiv.org/abs/1909.10300 )
> > > > >
> > > > > In summary, our paper contributes the following: (1) robustness bounds that are tighter than previous Lipschitz-based bounds, using the modulus of continuity; and (2) a fast & scaleable implementation of gradient regularization that empirically reduces the modulus of continuity in addition to the gradient, leading to adversarially models comparable to adversarial training. We show that gradient regularization is a valid method for promoting adversarial robustness. Prior to our work it was not clear if gradient regularization could compete with adversarial training. Overall we argue that these contributions alone are very valuable insights to the community.
> > > > >
> > > > > [2019-11-14 edited for TeX errors]

---

> > > > > ### Author Response · Authors · 2019-11-14
> > > > > **finite differences are well-studied**
> > > > >
> > > > > Hi, I think it's important to make a distinction between aspects of the paper that are new to this area, but are nevertheless well understood in general.  It's hard to fully explain finite differences in this venue, but they have been studied extensively. In particular, finite differences work well for a range of values: to large and they lose accuracy, too small, and they are vulnerable to rounding errors.  In this case, the performance of the model for h < 1e-3 may simply be due to rounding errors. Rounding errors are a well-studied phenomenon:  https://en.wikipedia.org/wiki/Round-off_error
> > > > >
> > > > > Keep in mind that symbolic differentiation (i.e. backpropagation) is one technique for estimating derivatives, and finite differences is anther.  Both of these are being combined with stochastic gradient descent training. For smooth models, the estimated derivatives should be robust to the method used.  The fact that the results of training deep models are so sensitive to the method of approximation used is itself motivation for our study of regularization/modulus of continuity.

---

> ### Author Response · Authors · 2019-11-14
> **Study on the effect of h**
>
> As requested, we have updated the paper with a study on the affect of $h$ on the performance.

---

### Official Review · AnonReviewer1 · 2019-10-24
**Official Blind Review #1**

**Rating:** 3

**Review:**

This paper proposed new regularizer for training robust models that can defend against evasion attack/adversarial examples. It looks to me there are two major novelties here, the authors suggest that 1) use dual norm on the gradient/jacobian as regularizer is tighter/better than the same norm or simply l2 norm; 2) the gradient/jacobian can be estimated by finite difference when moving the weight with a small step towards the gradient direction.

I review the paper with a standard for empirical paper. Please kindly clarify the theoretical contributions if the authors thought the theory is novel and important.

I donot think the empirical results are ready for publication. The final objective, when plugin finite difference into the norm regularizer, looks like logits squeezing https://openreview.net/forum?id=BJlr0j0ctX, or logits pairing https://arxiv.org/abs/1803.06373. Both methods are a little bit controversial.

There are several issues in the experiments. Several important baselines are missing. The paper did not compare with any of the regularizer-based robust models. When considering efficiency and fast training, the authors also did not compare with recent fast method Shafahi et al. 2019 Free and Zhang et al. 2019 YOPO.

When comparing with PGD adversarial training (Madry), in table 1, there is a more than 10% drop on robust accuracy for CIFAR-10 when \epsilon=8.

For l2 norm attack, how to interpret table 2? Why not provide accuracy under norm constraint like table 1?

Some uncommon settings are used in the experiments such as ResNeXt-34 and attacking randomly selected 1000 images.

Some relatively minor issues, could the authors elaborate on why the optimal value of max_v l( x+v) - c(v) is the squared dual norm of \grad l?

Typo in title: scaleable -> scalable


========= after rebuttal =============
I thank the authors for detailed replies. I still cannot support paper because
(1) the authors emphasize the theoretical contribution and claims the bound are tighter. However, they did not directly compare with any certified robust methods, or previous bounds to support the argument.
(2) The empirical results look suboptimal. The authors did not convince me why they sampled 1000 images for test for a small CIFAR-10 dataset. The proposed method is 10% less robust comparing to Madry's in table 1.

**Experience Assessment:**

I have published one or two papers in this area.

**Review Assessment: Checking Correctness Of Derivations And Theory:**

I assessed the sensibility of the derivations and theory.

**Review Assessment: Checking Correctness Of Experiments:**

I assessed the sensibility of the experiments.

**Review Assessment: Thoroughness In Paper Reading:**

I read the paper thoroughly.

---

> ### Author Response · Authors · 2019-11-09
> **Reply to reviewer 1**
>
> Thanks for your review, we appreciate you taking the time to review our paper. We will address your criticisms in order.
>
> The work is a theory paper deriving robustness bounds which justify gradient regularization as a method for training adversarially robust models. It is not ‘just’ an empirical work; the empirical results we present in the last two pages backup our theoretical results. We are not claiming that gradient regularization is superior to adversarial  training. However our empirical results show that gradient regularization can achieve comparable robustness to adversarial training, at a fraction of the training time of ‘traditional’ adversarial training (in the style of say Madry et al).
>
> We don’t see the connection to logit squeezing. We are unaware of a reason why small logits should imply loss gradients are small, or vice versa.
>
> Our implementation of gradient regularization with finite differences indeed has some similarities to logit pairing. Logit pairing was proposed with a heuristic justification and lacked theoretical robustness guarantees. Our method is backed theoretically via robustness bounds. Our method could be seen as complimentary to logit pairing, in that for a special choice of the pairing function (difference of the training loss, which is not discussed in the original logit pairing paper), the implementations of logit pairing and finite difference gradient regularization coincide. Our work could  then be interpreted as providing some theoretical justification for logit pairing, in certain scernarios.
>
> As requested, we have amended our submission to include a comparison with ‘For Free’ adversarial training (Shafahi et al 2019), using the same hyperparameters as in that work. Please see the revised Table 1. Currently we have updated CIFAR-10; ImageNet-1k results will take awhile, and could be included later. We found that `For Free’ adversarial training was not able to outperform traditional adversarial training, nor did it outperform gradient regularization. It is very fast though!
>
> We did not compare against regularizer-based robust models because they are not the current state-of-the-art. Do you have a particular citation against which we should compare?
>
> Please take a look at the empirical results as a whole: our aim was not to outperform adversarial training in all circumstances. Instead we have proposed a method that is considerably faster, and which provides good robustness that is on the same order as the robustness of adversarial training. In many circumstances we would argue our approach is very desirable, for example when training is limited by time constraints.
>
> Regarding Table 2. Is it that you would prefer results reported at certain distances in L2? We chose instead to report the median distance, because (unlike in L-infinity) there is no commonly accepted distances to report robustness results at. Until the community agrees on a standard L2 distance for reporting, we think reporting the median distance is a fair metric. We did not want to be accused of cherry-picking our L2 results by selecting arbitrary L2 thresholds.
>
> There are many adversarial robustness papers which report results on only a subset of test images. It is very computationally demanding to obtain results on all test images.
>
> In our experience we have found ResNeXt outperforms ResNet and Wide-ResNets in terms of both robustness and test accuracy, and is faster to train. It is entirely fair to use newer & better architectures as they are developed. We don’t believe researchers should be constrained to use a particular architecture for reporting results.
>
> We don’t claim that max_v l( x+v) - c(v) is the squared dual norm of \grad l. However it is approximately equivalent (up to second order) when c(v) is the squared norm and l is differentiable. To see this, Taylor expand l(x+v); and discard second order terms. Then use Legendre duality to solve the (approximate) maximization problem directly. There is a derivation in Amir Beck’s book “First Order Methods in Optimization” I believe.
>
> About the spelling of ‘scaleable’: it depends which side of the Atlantic your grammatical allegiances lie :-) ... we used British English spelling.
>
> Thanks again for your review. We hope that this reply is helpful and provides some additional context to our paper.

---

> > ### Comment · AnonReviewer1 · 2019-11-13
> > **More clarification**
> >
> > Thanks for response. I could read the paper again later. I just want to give the authors another chance to clarify before rebuttal period ends.
> >
> > 1) Since the authors argue theory is the main contribution of the paper.  I would like the authors to clarify what the theory tells us. Could the bound be used in anyway? Why is the bound novel? Since so many approximation has been made, what is the direct connection between bound and the proposed method?
> >
> > 2) The response on efficiency and robustness looks confusing to me. The authors first claim Free is as good as PGD, but not as good as the proposed method. Later say the purpose of the proposed method is not for better robustness accuracy so it is fine to have 10% drop on robustness accuracy comparing to PGD. The proposed method does not reach state-of-the-art, why not compare with other input regularization method, just for your own good to show advantages. An example of previous method is "Improving the Adversarial Robustness and Interpretability of Deep Neural Networks by Regularizing Their Input Gradients", there are several more recent works.
> >
> > 3) I am curious to get a few reference on using sampled validation set to evaluate robustness for CIFAR.

---

> > > ### Author Response · Authors · 2019-11-14
> > > **Clarification**
> > >
> > > Thanks for the timely reply. Regarding your remarks:
> > >
> > > 1) We provide two bounds which say that robust models must necessarily have small gradients and small moduli of continuity.  The (w)-bound says that if both the gradients of the loss are small and the modulus of continuity is small (and the model has small loss), the network will be adversarially robust up to perturbations of a certain size. The (L)-bound says if the Lipschitz constant of the loss is small, the network is robust up to perturbations of a certain size. These bounds are valid in any norm.
> > >
> > > In practice we use these bounds in Tables 2 & 3 for the L-infinity and L2 norms. Each table has a column with the predicted number of perturbed images at a particular threshold (using the theoretical bounds), as well as the usual columns showing empirically how many images were perturbed. We can see that the theoretical bounds do a good job of capturing the robustness of the model, without having to actually attack the model.
> > >
> > > As for novelty. These bounds are tighter and more accurate than existing ones in the literature. To our knowledge, all bounds to date have used the model outputs, whereas ours uses the loss of the model.  Moreover, the literature has focused on the Lipschitz constant of the model for deriving bounds. We have introduced the importance of the modulus of continuity in adversarial robustness bounds.  The modulus of continuity provides bounds that are locally more accurate than Lipschitz-type bounds. This is because they use two local pieces of information, the loss value and the gradient, whereas Lipschitz-type bounds use only one local piece of information.
> > >
> > > The proposed method, gradient regularization with finite differences, is directly motivated by (w)-bound: the method ensures that both gradients are small and the modulus of continuity is small. See for example Table 3 where we empirically demonstrate that gradient regularization with finite differences ensures small gradients and moduli of continuity.
> > >
> > > The only approximations we have made are (i) when we estimate the modulus of continuity and the Lipschitz constant for estimating the bounds, using techniques from Extreme Value Theory (as in Weng et al, ICLR 2018); and (ii) finite differences. The former does not play a role in the proposed training method; the approximation is only used after-the-fact on trained models. Finite differences are important for controlling the modulus of continuity: compare gradient regularization via autograd versus with finite differences. Furthermore, finite differences are vastly faster than double backpropagation.
> > >
> > > 2) There are two measures here: efficiency, and robustness. Let’s compare: (a) AT, Madry-style; (b) AT, “For Free”; and (c)  our proposed method. If the end user only cares about robustness, then the ranking is: (1) Madry-style AT,  (2) our method, (3) AT “For Free”. However note that our method performs nearly as well as Madry-style AT: refer to Figure 1, where we show that the robustness curves of our method are comparable to Madry-style AT. If instead the end user cares about training time, the ranking is (1) AT “For Free”; (2) our method; (3) Madry-style AT.
> > >
> > > Our proposed method achieves nearly state-of-the-art at fractions of the training time in L-infinity. In L2 the proposed method surpasses adversarial training.
> > >
> > > We have updated the tables to compare against double backpropagation, the method used in “Improving Adversarial Robustness ... by Regularizing their Gradients”  (Ros & Doshi-Velez, AAAI 2018). As can be seen in Table 2, our method outperforms double backpropagation.
> > > Moreover, Ros & Doshi-Velez only studied robustness in L2, only compared against one-step PGD attacks, and only compared against one-step adversarial training. Thus in our opinion it wasn’t clear from that work whether gradient regularization could compete with adversarial training. We believe our work answers this question.
> > >
> > > 3) It’s commonly accepted that statistics can be estimated from a sample of the dataset. We randomly chose 1000 of the 10000 validation images, which should be enough statistically.
> > >
> > > Many other adversarial robustness papers have done this. To name a few: “Decision-Based Adversarial Attacks: Reliable Attacks Against Black-Box Machine Learning Models” (Brendel et al, ICLR 2018); “Certified Adversarial Robustness via Randomized Smoothing” (Cohen et al, ICML 2019); “Evaluating the robustness of neural networks: An extreme value theory approach” (Weng et al, ICLR 2018); and “Differentiable abstract interpretation for provably robust neural networks” (Mirman et al, ICML 2018).

---

### Author Response · Authors · 2019-11-09
**General reply to reviewers**

We thank all the reviewers for their comments. We’d like to emphasize what we believe are the main contributions of this work.

First, we derive theoretical robustness bounds. In practice the bounds are fast to evaluate on a per-image basis, and which serve as a fairly accurate heuristic assessing vulnerability to adversarial attacks. Moreover the bounds explain why small gradients are not sufficient alone for adversarial robustness, in that the modulus of continuity of the gradients can still be quite large.

Second, these bounds justify training with gradient regularization, implemented using finite differences. Not only does this form of gradient regularization yield comparable robustness to adversarial training (on the same order), it is also very fast, and does not unduly impede training time.

---

### Decision · Program_Chairs · 2019-12-19

**Decision:**

Reject

**Comment:**

(1) the authors emphasize the theoretical contribution and claims the bound are tighter. However, they did not directly compare with any certified robust methods, or previous bounds to support the argument.
HM, not sure, need to check this

(2) The empirical results look suboptimal. The authors did not convince me why they sampled 1000 images for test for a small CIFAR-10 dataset. The proposed method is 10% less robust comparing to Madry's in table 1.
Seems ok, understand authors response

1) The theoretical analysis are not terribly new, which is just a straightforward application of first-order Taylor expansion. This idea could be traced back to the very first paper on adversarial examples FGSM (Goodfellow et al 2014).
True

2) The novelty of the paper is to replace exact gradient (w.r.t input) by their finite difference and use it as a regularization. However, there is a misalignment between the theory and the proposed algorithm. The theory only encourages input gradient regularization, regardless to how it is evaluated, and previous studies have shown that this is not a very effective way to improve robustness. According to the experiments, the main empirical improvement comes from the finite difference implementation but the benefit of finite difference is not justified/discussed by the theory. Therefore, the empirical improvement are not supported by the theory. Authors have briefly respond to this issue in the discussion but I believe a more rigorous analysis is needed.
This seems okay based on author response

3) Moreover, the empirical performance does not achieve state-of-the-art result. Indeed, there is a non-negligible  gap (12%) between the obtained performance and some well-known baseline. Thus the empirical contribution is also limited.
Yea, for some cases